# Working from Home in Italy during COVID-19 Lockdown: A Survey to Assess the Indoor Environmental Quality and Productivity

Francesco Salamone *, Benedetta Barozzi, Alice Bellazzi, Lorenzo Belussi, Ludovico Danza, Anna Devitofrancesco, Matteo Ghellere, Italo Meroni, Fabio Scamoni and Chiara Scrosati

Construction Technologies Institute, National Research Council of Italy (ITC-CNR), Via Lombardia, 49, San Giuliano Milanese, 20098 Milan, Italy; barozzi@itc.cnr.it (B.B.); bellazzi@itc.cnr.it (A.B.); belussi@itc.cnr.it (L.B.); danza@itc.cnr.it (L.D.); devitofrancesco@itc.cnr.it (A.D.); ghellere@itc.cnr.it (M.G.); meroni@itc.cnr.it (I.M.); scamoni@itc.cnr.it (F.S.); scrosati@itc.cnr.it (C.S.)
* Correspondence: salamone@itc.cnr.it

**Abstract:** Italians were the first European citizens to experience the lockdown due to Sars-Cov-2 in March 2020. Most employees were forced to work from home. People suddenly had to share common living spaces with family members for longer periods of time and convert home spaces into workplaces. This inevitably had a subjective impact on the perception, satisfaction and preference of indoor environmental quality and work productivity. A web-based survey was designed and administered to Italian employees to determine how they perceived the indoor environmental quality of residential spaces when Working From Home (WFH) and to investigate the relationship between different aspects of users' satisfaction. A total of 330 valid questionnaires were collected and analysed. The article reports the results of the analyses conducted using a descriptive approach and predictive models to quantify comfort in living spaces when WFH, focusing on respondents' satisfaction. Most of them were satisfied with the indoor environmental conditions (89% as the sum of "very satisfied" and "satisfied" responses for thermal comfort, 74% for visual comfort, 68% for acoustic quality and 81% for indoor air quality), while the layout of the furniture negatively influenced the WFH experience: 45% of the participants expressed an unsatisfactory or neutral opinion. The results of the sentiment analysis confirmed this trend. Among the Indoor Environmental factors that affect productivity, visual comfort is the most relevant variable. As for the predictive approach using machine learning, the Support Vector Machine classifier performed best in predicting overall satisfaction.

**Keywords:** working from home; survey; questionnaire; indoor environmental quality; COVID-19 lockdown; productivity

## 1. Introduction

Over the years, words such as "agile working" and "flexible working" have been used frequently. Since the early 2020s, the word "working" has become a neologism in conjunction with some specific adjectives from different countries to denote a specific situation. For example, "Working From Home" (WFH) is used to describe a work activity carried out far from the office, mainly in English-speaking countries. The definition of "smart working" draws on the positive meaning commonly attached to this adjective, especially in Italy, and is used in this context to emphasise high efficiency in achieving work results through a combination of flexibility, autonomy and collaboration, in parallel with the optimisation of tools and working environments for employees [1]. Instead, the word "working" is used in conjunction with each of these adjectives (i.e., "smart", "flexible" or "agile") to highlight certain characteristics. For example, while "remote working" (or WFH) is used to describe work activity performed far from the office, "flexible" is used in a spatio-temporal sense to emphasise the flexibility of work that is compatible

with family life. The term "agile working" is used to define "a set of practises that enable organisations to build an optimal workforce and provide the benefits of a greater match between resources and demand for services, increased productivity, and improved talent attraction and retention" [1]. Finally, the definition of "smart working" draws on the positive meaning commonly attributed to this adjective and is used in this context to emphasise high efficiency in achieving work outcomes through a combination of flexibility, autonomy and collaboration, in parallel with optimising tools and work environments for employees [2]. The way people worked during the lockdown can be considered as WFH, because they managed various difficulties due to the sudden closure of the offices. The aim of this study is to analyse the Indoor Environmental Quality (IEQ) of home offices and the productivity of workers during the Coronavirus pandemic. To this end, a web-based survey was designed and submitted to Italian workers to assess how they perceive the indoor quality of home spaces during WFH and to explore the relationship between different aspects of user satisfaction. The survey was conducted during the COVID-19 lockdown in Italy, when workers were unexpectedly forced to work at home.

In this context, Table 1 of Section 1.1 lists the papers dealing with satisfaction during WFH, while Table 2 in Section 1.2 lists the manuscripts dealing with the use of surveys to assess IEQ.

### 1.1. Reference Literature on the Topic of Working for Home

In recent decades, several researchers have studied the impact of WFH on worker productivity and well-being. Some of them show how people who work from home can suffer from stressful consequences [3,4], a kind of disorder in WFH, due to social and professional isolation, difficulties in communication, and changes in social relationships due to the absence of face-to-face interaction [5]. In contrast, many studies have shown a positive relationship between worker satisfaction and WFH [6,7]. This seems reasonable considering that some benefits closely associated with WFH, such as time and money saving, could have a positive impact on all workers [8], provided that workers activating WFH for the first time and those already experienced with teleworking develop comparable levels of job satisfaction [9]. These studies are described in Table 1 and mainly refer to situations where WFH is an Experimental (E) or Consolidated (C) practise, namely a structured solution created by the company as an alternative to coworking. There is a lack of studies on satisfaction in an Adaptive (A) situation related to emergencies. This is what happened in Italy, where WFH was hardly used [10] when, at the end of February 2020 the Italian Government faced the Sars-Cov-2 pandemic, and the introduction of WFH was one of the measures taken to contain the epidemiological emergency. It was therefore an emergency measure that did not allow this form of work to be organised in a structured way: Workers suddenly found themselves in a work context for which they were psychologically and organisationally unprepared, and most workers had to adapt their domestic spaces and technologies to home working [11].

The novelty of the current research is to investigate the IEQ of workers during WFH activities that, to the best authors' knowledge, has not been sufficiently investigated before. The outcome of this study can be the basis for the formulation of strategies to improve WFH and, in particular, the indoor conditions of living spaces where work activities are performed.

Table 1. Main features of the reference literature on the topic of Working From Home. WFH typology: C—Consolidated, E—Experimental, A—Adaptive.

| Ref. | WFH Typology | Number of Participants | Research Type | Category of the Assessment | Methodological Approach | Results |
|---|---|---|---|---|---|---|
| [3] | C | 258 teleworkers from two global Information Technology companies | Predictive | Job satisfaction according to technostress indicators | Survey | Teleworking produces technostress due to the job characteristics and technology as a function of the intensity of teleworking: The higher the intensity, the less technostress |
| [4] | C/E | 37,553 teleworkers (from 63 different studies) | Review | Affective, cognitive, social, professional and psychosomatic. | - | Remote working has a positive effect on job satisfaction, organisational commitment, reduction of emotional exhaustion and working autonomy and a negative effect due to social isolation and carrier advancement |
| [5] | C | N.A. (34 studies) | Review | Impact on work life and job satisfaction from four perspectives: communication, social relation, achievement recognition and work-life balance. | - | Remote working produces difficulties in communication, reduction of social relationship, achievement recognition and work-life balance |
| [6] | C | 45,000 workers (from Labour Force Surveys) & 4470 (2001) 7787 (2006) and 3200 (2012) workers | Review/theoretically informed predictions | Work effort, job- related well- being and work- life balance. | Surveys | Remote working has a positive effect on employing organization, commitment and job satisfaction and a negative effect on work-life balance |
| [7] | C | 180 | Descriptive | Job performance, satisfaction and creativity | Surveys | Higher levels of job performance, job satisfaction and creative performance in WFH. |
| [8] | C | 7500 | Descriptive | Relationship between WFH and extra work activities depending on the type of employee: how they use the saved travel time and budget | Survey | Telecommuting is associated with more out-of-home activities, and it improves the workers' work–life balance |

*1.2. Survey Definition Based on the Reference Literature*

The use of questionnaires to determine satisfaction and comfort in the work environment is widespread. Huizenga et al. [12] emphasise that the user is the most important source of information for understanding the overall performance of a building. For this reason, the authors developed, tested and refined a survey based on a standardised set of questions to assess occupant satisfaction in relation to various environmental factors (Indoor Air Quality—IAQ, Thermal Comfort—TC, Acoustic Quality—AQ). Several studies have shown that satisfaction is closely correlated with office layout. Kwon et al. [13], for example, report the results of a survey of 579 office users. The aim was to investigate and identify the most influential design factors on user satisfaction. The location of the desk and the layout of the office were found to have a major influence on TC and Visual Comfort (VC). Candido et al. [14] describe the results of the BOSSA project (Building Occupants Survey System Australia), which aims to develop and implement tools to study overall IEQ performance in office buildings in Australia. The BOSSA Time-Lapse IEQ questionnaire was submitted by more than 5000 office workers. The results of the survey show an increase in satisfaction and an improvement in employees' working lives with better office facilities. Graham et al. [15] report analysis results from two decades of data collected by the Center for the Built Environment (CBE) Occupant Survey. The results show that the ease of interaction in spaces, the amount of light and cleanliness are the main reasons for satisfaction, while sound privacy, temperature and noise levels are the main reasons for dissatisfaction. Frontczak et al. [16], upon examining ten years of collected CBE data in a US office building, identify the amount of space and visual privacy as the most important parameters for classifying a workplace as satisfactory. Wong et al. in [17], define a multivariate logistic regression model for IEQ considering 293 workers' acceptance feedbacks on four aspects (thermal comfort, indoor air quality, noise level and lighting level) in different offices in Hong Kong. The results show that all aspects are important for IEQ acceptance, but with different weighting: Thermal environment has the highest relative importance, followed by air quality, noise level and lighting level. Choi et al. in [18], state that worker satisfaction is also influenced by the control over indoor climate conditions, the level of privacy, office space and furnishings.

T. Crosbie and J. Moore [19] note that while WFH offers the potential for greater flexibility in non-pandemic times, it does not facilitate the dismantling of traditional gender roles. Rupietta et al. [20] find that WFH has a statistically significant positive impact on work effort and also has a positive impact on unpaid overtime.

Shareena and Shahid [21] demonstrated in the Covid-19 pandemic that willingness to engage in WFH depends entirely on having children at home, a comfortable room, a quiet environment and good internet access. These aspects are the most important variables, along with physical activity, communication with colleagues, workplace layout and indoor environment satisfaction as highlighted by Xiao et al. [22].

Like previous works, this study is also based on surveys. What is new is that the scope is extended to home spaces used for working activities, where people have the possibility to adapt their conditions to achieve high levels of satisfaction. The aim is to identify the variables that can influence the satisfaction of WFH users. Table 2 summarises the main characteristics of the reference literature.

**Table 2.** Survey: Main characteristics of the reference literature.

| Ref. | Case | Number of Participants | Research Type | Category of the Assessment | Köppen-Geiger Climate Classification [20] (Period) | Results |
|---|---|---|---|---|---|---|
| [12] | 22 buildings in the United States, including offices, laboratories, banks and courthouses | - | Causal explanatory | Satisfaction with indoor environments | Case 1: BSk, Dfa (summer 2001); Case 2: Csb, Bsk, Csa(November 2001) Case 3: BSk, Dfa (-), Csb, Bsk, Csa(-), Dfa, Dfb (-), Cfa (-) | Average satisfaction follows this ranking: 1. Lighting, 2. Layout, 3. Building and ground, 4. Wayfinding, 5. Air quality, 6. Thermal comfort and acoustic quality (same rating) |
| [13] | Five office cases in the Netherlands | 579 | Descriptive | Influential office design factors on occupant's satisfaction | Cfb(different seasons) | Desk location and office layout have a large impact on occupant's satisfaction |
| [14] | 30 office buildings from the BOSSA database | 5171 | Descriptive | Impact of different workspace layouts on occupants' overall satisfaction on key IEQ dimensions, perceived productivity and perceived health | BWh, BWk, BSh, Csa, Csb, Cfa, Cfb, (different seasons) | Occupant's satisfaction is related to the workspace layout |
| [15] | ~900 buildings by CBE database including office, education spaces, laboratories, healthcare workspaces, multi-family residential dormitories | >90,000 | Descriptive | Satisfaction with indoor environmental factors, privacy, furniture | Worldwide (different seasons) | Spaces' ease of interaction, amount of light and cleanliness are the main source of satisfaction. Sound privacy, temperature and noise level are the main causes of dissatisfaction. |
| [16] | 351 office buildings by CBE database | 52,980 | Descriptive | Satisfaction with indoor environmental parameters, workspace, and building features | US (different seasons) | The most important parameters were satisfaction with the amount of space, noise level and visual privacy. |
| [17] | 3 different types of offices in Hong Kong | 293 | Descriptive/ predictive | Acceptance of the perceived indoor environment is given by four aspects, namely thermal environment, indoor air quality, equivalent noise level and illumination level. | Cfa (-) | Operative temperature is the most influential variable. |

**Table 2.** *Cont.*

| Ref. | Case | Number of Participants | Research Type | Category of the Assessment | Köppen-Geiger Climate Classification [20] (Period) | Results |
|---|---|---|---|---|---|---|
| [18] | 29 office buildings in US | 492 | Descriptive | Satisfaction with the indoor environments including thermal, air, acoustic, lighting, spatial and overall environmental qualities. | - | Each sub-set of IEQ satisfaction is ranked using statistical methods to illustrate respective contribution to overall environmental satisfaction, given workstation location and occupant gender. |
| [19] | Six of the interviews took place in the participants' homes, the focus groups in surroundings familiar to the participants | 45 participants, 3 focus groups | Descriptive | Balancing between WFH and work-life | Csa | Positive parameters are flexibility and choice for a broad range of workers. Negative: doesn't facilitate the breakdown of traditional gender roles nor does it smooth out the inequalities that education and income bring. |
| [20] | SOEP dataset concerning German employees WFH | 22,000 | Descriptive | Satisfaction with indoor environmental factors, privacy, furniture | Cfb, Dfb | Indoor environmental factors, privacy and furniture are statistically significant on work effort and on unpaid overtime hours |
| [21] | 50 employees from IT sectors to teaching sectors in Mangalore (India) | 50 | Descriptive | Productivity and satisfaction in WFH versus working at office | Am | Presence of children at home, comfortable space at home, quiet environment at home and good internet access are the most important variables |
| [22] | Participants were recruited through emails, social media platforms and newsletters in California | 988 valid questionnaires | Descriptive | Social, behavioural and physical factors on well-being of office workstation users during COVID-19 WFH | Csa, Csb | Physical exercise, communication with colleagues, children at home, distractions while working, workstation set-up and satisfaction with indoor environmental factors |

### 1.3. Motivation and Goals

The Google Trends service provides accurate and representative information about users' online search habits [23], especially during the pandemic, when the Italian government closed down all non-essential activities such as shops (except grocery stores), offices, restaurants, cinemas, gyms, etc., and the entire public administration sector and most private companies were forced to WFH to avoid any possibility of spreading the virus. Figure 1 shows that at the end of February 2020, when the pandemic emergency hit the country hard, the word "smart working", often used in Italy to refer to WFH, and other keywords related to office furniture (keyboard, printer, headphones and notebook) reached a peak compared to the trend of the previous year. This underlines the increasing demand for necessary equipment (Figure 1). It was created using the pytrends library application considering a time frame between 15 April 2019 and 15 October 2020, geo = "IT", no group, no specific category. For more details, see [24].

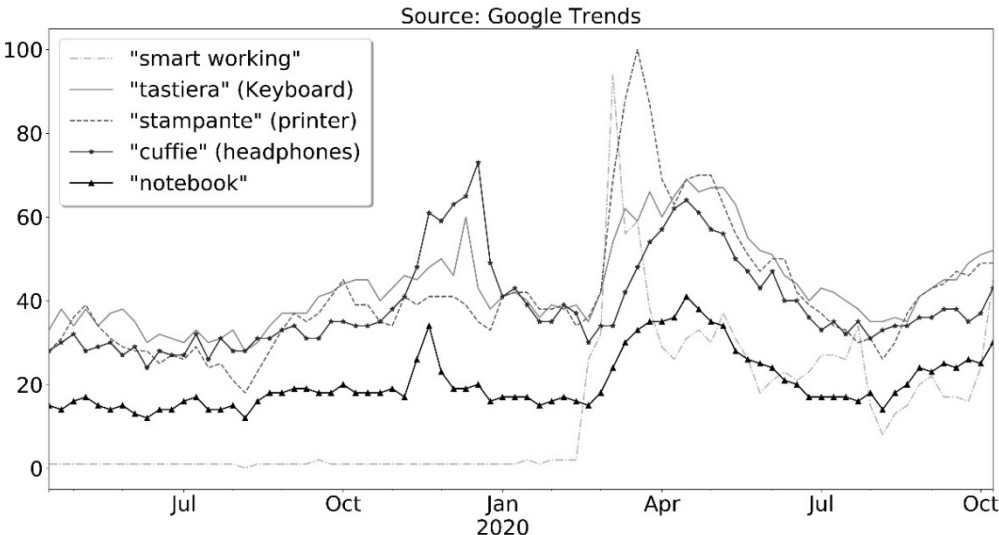

**Figure 1.** Google Trends in the last years in Italy: Comparison among "smart working" a keyword usually used in Italy to identify Working From Home, and other equipment related words (in Italian, the meaning in brackets).

As a result of blocking all non-essential commercial activities, online demand has increased.

In most cases, houses in Italy are dwellings in multifamily buildings or building blocks with an area of about 90 m$^2$, including a kitchen and a living room, one or two bathrooms and two bedrooms [25]. Consequently, work activities have to be carried out in rooms that are not intended for this purpose. As shown by Eurostat data in 2019, only 1.1% of all workers aged 15–64 have occasionally adopted a WFH solution [26], indicating that WFH is not a widespread and consolidated reality in Italy. It follows that most workers were not yet ready to switch from a co-worker mode to WFH.

Furthermore, the lockdown has significantly changed people's lifestyles, as the restriction of movement has forced entire families to live together during working days.

In this new perspective, homes should meet the requirements for hosting work activities: From the availability of equipment to adequate spaces, from ergonomics to indoor environmental quality. However, today the question is: "are houses ready for the new lifestyle?". Specifically, this study pursues the following goals, which differ in the analytical approach chosen and are each linked to a specific question:

- Goal 1—Descriptive approach. Question 1: How do home workers feel the IEQ during working activities at home? Question 2: Are they satisfied about their productivity?
- Goal 2—Predictive model. Question 3: What are the most suitable algorithms and influential features to predict workers' overall satisfaction?

- Goal 3—Sentiment analysis. Question 4: What impact does WFH have on workers' satisfaction?

In particular, the information collected through the questionnaire is analysed to determine users' perception, preference and satisfaction in relation to TC and VC, AQ and noise level, IAQ and overall comfort. Data on users' satisfaction are the basis for the implementation of a Machine Learning (ML) algorithm that aims to predict IEQ while WFH. Finally, sentiment analysis, a field of Natural Language Processing (NLP) [27] used to interpret and classify emotions in subjective data is performed to understand the impact of WFH on productivity levels.

## 2. Method

The research method followed in this study is based on three main steps: Data acquisition, data preparation and data analysis.

### 2.1. Data Acquisition

Participants' personal feedback on their comfort and productivity was collected using a web-based survey on the Microsoft Forms platform in compliance with the European General Data Protection Regulation GDPR UE 2016/679. The structure of the survey, defined in accordance with [28], consists of seven main sections:

- Section A: Each participant must give consent for the use of the given answers for scientific purposes, according to the guidelines of GDPR UE 2016/679;
- Section B: Personal information defining the users and their household;
- Section C: Information on the dwelling characteristics, such as building type (detached house or apartment), year of construction and refurbishment interventions;
- Section D: Characteristics of the rooms commonly used in WFH, such as dimensions and layout, time spent in the room, appliances and controls and ease of use;
- Section E: Information of single aspects of comfort in the following order TC, VC, AQ and IAQ. Each aspect is investigated from three perspectives: Satisfaction, preference and perception. For each category, a specific Categorical Scale (CS) [29] is defined with different points (see Table 3). Satisfaction is analysed with a 5-point scale, ranging from "Very dissatisfied" (−2) to "Very satisfied" (+2). For the question on preference, the bipolar 3-point scale is used, except for acoustic quality that requires one polar 4-point scale. The perception category differs from aspect to aspect. TC uses the 7-point scale of the ASHRAE standard [30], VC a bipolar 5-point scale, AQ a unipolar 5-point scale and, finally, IAQ a unipolar 4-point scale. The interference of all the considered environmental factors in relation to the productivity of the work activity is defined by considering a bipolar 3-point scale. Figure 2 shows, as an example, the format used and relative questions (translated into English) related to TC;
- Section F refers to overall satisfaction and satisfaction in relation to some other aspects that may affect working activities: Available space, furniture and accessories, visual and acoustic privacy, productivity. Participants have to answer questions about the relative importance of ten specific aspects in relation to satisfaction and how WFH could affect the comfort perception. Table 3 summarises the characteristics of the scales used to identify the satisfaction, preference and perception of participants in relation to the considered aspects and the relative scores. Depending on the scale used (one polar or bipolar), the value zero is not always associated with a "neutral" score. With bipolar scales, the neutral value zero is the best condition. However, with AQ preference, AQ perception and IAQ perception characterised by one polar scale, the value zero is associated with the best condition;
- Section G provides an open-ended question that allows participants to give their comments.

Table 3. Categorical Scale (CS) used in the questionnaire and relative scores.

| Parameter | Scale | Vote | −3 | −2 | −1 | 0 | 1 | 2 | 3 | 4 |
|---|---|---|---|---|---|---|---|---|---|---|
| TC | 5-point CS | Satisfaction | - | Very dissatisfied | Dissatisfied | Neither satisfied nor dissatisfied | Satisfied | Very satisfied | - | - |
| | 3-point CS | Preference | - | - | Cooler | No change | Warmer | - | - | - |
| | 7-point CS | Perception | Cold | Cool | Slightly cool | Neutral | Slightly warm | Warm | Hot | - |
| | 3-point CS | Interference | - | - | Interfere | Neutral | Enhance | - | - | - |
| VC | 5-point CS | Satisfaction | - | Very dissatisfied | Satisfied | Neither satisfied nor dissatisfied | Satisfied | Very satisfied | - | - |
| | 3-point CS | Preference | - | - | Darker | No change | More light | - | - | - |
| | 5-point CS | Perception | | Dark | Slightly dark | Neutral | Slightly bright | Bright | - | - |
| | 3-point CS | Interference | - | - | Interfere | Neutral | Enhance | - | - | - |
| AQ | 5-point CS | Satisfaction | - | Very dissatisfied | Dissatisfied | Neither satisfied nor dissatisfied | Satisfied | Very satisfied | - | - |
| | 4-point CS | Preference | | | | No change | Slightly quieter | Quieter | Much quieter | |
| | 5-point CS | Perception | | | | Very quiet | Quiet | Slightly Noisy | Noisy | Very noisy |
| | 3-point CS | Interference | - | - | Interfere | Neutral | Enhance | - | - | - |
| IAQ | 5-point CS | Satisfaction | - | Very dissatisfied | Dissatisfied | Neither satisfied nor dissatisfied | Satisfied | Very satisfied | - | - |
| | 4-point CS | Perception | | | | Not smelly | Slightly smelly | Smelly | Very smelly | - |
| | 3-point CS | Interference | - | - | Interfere | Neutral | Enhance | - | - | - |
| Overall satisfaction | 5-point CS | Satisfaction | - | Very dissatisfied | Dissatisfied | Neither satisfied nor dissatisfied | Satisfied | Very satisfied | - | - |
| Amount of space | 5-point CS | Satisfaction | - | Very dissatisfied | Dissatisfied | Neither satisfied nor dissatisfied | Satisfied | Very satisfied | - | - |
| Comfort of furniture | 5-point CS | Satisfaction | - | Very dissatisfied | Dissatisfied | Neither satisfied nor dissatisfied | Satisfied | Very satisfied | - | - |
| Visual privacy | 5-point CS | Satisfaction | - | Very dissatisfied | Dissatisfied | Neither satisfied nor dissatisfied | Satisfied | Very satisfied | - | - |
| Acoustic privacy | 5-point CS | Satisfaction | - | Very dissatisfied | Dissatisfied | Neither satisfied nor dissatisfied | Satisfied | Very satisfied | - | - |
| Productivity | 5-point CS | Satisfaction | - | Very dissatisfied | Dissatisfied | Neither satisfied nor dissatisfied | Satisfied | Very satisfied | - | - |

## Thermal comfort

Are you satisfied with the temperature of the environment? *

| Very satisfied | Satisfied | Neither satisfied nor dissatisfied | Dissatisfied | Very dissatisfied |
|:---:|:---:|:---:|:---:|:---:|
| ○ | ○ | ○ | ○ | ○ |

What would you like the thermal environment to be like? *

| Warmer | No change | Cooler |
|:---:|:---:|:---:|
| ○ | ○ | ○ |

How do you perceive the thermal environment? *

| Cold | Cool | Slighly cool | Neutral | Slightly warm | Warm | Hot |
|:---:|:---:|:---:|:---:|:---:|:---:|:---:|
| ○ | ○ | ○ | ○ | ○ | ○ | ○ |

How does the thermal comfort of the environment interfere with your ability to work? *

| Ehnance | Neither enhance nor interfere | Interfere |
|:---:|:---:|:---:|
| ○ | ○ | ○ |

**Figure 2.** Satisfaction, preference, perception and interference scales related to Thermal Comfort. The "*" identify mandatory questions.

The questionnaire was structured to better understand the potential impact of the sudden massive use of WFH on people, which can be explained by considering the following influential factors:

- The type of dwelling and rooms worked in;
- The prevalence of WFH in Italy before the lockdown;
- The imposed conditions of "living together" and the possible presence of children during working hours due to closed schools and distance learning.

The questionnaire was made available on the ITC-CNR website [28] and distributed to CNR staff and civilian employees during the lockdown period, from April to June 2020. The survey was open to all interested parties.

### 2.2. Data Preparation

The questionnaire was distributed to civil servants, including university staff and private employees, who mainly perform office work, because in this case, working from home is equivalent to working in an office. Civil servants are a broad category of workers

and had to work from home during lockdown. Similarly, many private employees were forced to work from home. A total of 330 respondents from all over Italy, which is characterised by a predominantly temperate climate [31], gave their consent and participated in the survey over a period of 3 months from April to June 2020, during spring when the air temperature at 2 m altitude was not lower than 8 °C and not higher than 25 °C, as indicated in Italian Air Force technical reports [32–34]. Participants were asked to answer the questionnaire once during the survey. All 330 participants' feedback was verified and considered valid for the analysis. Most of the columns in the response database are of object type, i.e., they contain strings. These variables are categorical and are stored as text values representing different aspects: For the question related to the environment where WFH takes place, participants are asked to answer considering strings such as "Kitchen", "Living room", "Bedroom", "Study room" and "Other" (to be specified). The filtering phase avoids redundancies among the users' answers. For this reason, all considered features are counted and aggregated into a clear set of possible outputs. In the encoding phase, the aggregated answers are converted into numerical values for further processing, since many Machine Learning (ML) algorithms can support categorical labels without further modifications, but many others cannot. Table 4 shows an example of the encoding phase performed for the answers of the Env_SM feature used to identify where the WFH was performed.

**Table 4.** Provided answers (object data type) and considered answer (int64 data type) to question: Identify the room where you perform WFH (feature Env_SM).

| Answer Provided by Participants (in Italian in Quotes, the Meaning on the Right) | | Number | Aggregated Answer (Encoding) | Number |
|---|---|---|---|---|
| *"Studio"* | Study room | 83 | Study room (1) | 83 |
| *"Soggiorno"* | Living Room | 129 | Living room (0) | 132 |
| *"Angolo studio in soggiorno"* | Study corner within living room | 1 | | |
| *"Living room"* | Living room | 1 | | |
| *"Salotto"* | Living room | 1 | | |
| *"Mi sposto nei vari locali (cucina/soggiorno/camera)"* | Not fixed room (Kitchen/Living room/Bedroom) | 1 | Moving (6) | 4 |
| *"Cucina, camera, soggiorno"* | Kitchen, Bedroom, Living room | 1 | | |
| *"Dove capita"* | Wherever | 1 | | |
| *"Cucina oppure soggiorno oppure cameretta"* | Kitchen or Living room or Bedroom | 1 | | |
| *"Cucina"* | Kitchen | 46 | Kitchen (2) | 46 |
| *"Camera da letto"* | Bedroom | 46 | Bedroom (3) | 48 |
| *"Camera figlio"* | Children's room | 1 | | |
| *"Stanza per ospiti"* | Guest room | 1 | | |
| *"Soggiorno/cucina open space"* | Living/Kitchen in open space | 2 | Living area (single room) (5) | 9 |
| *"soggiorno-cucina unica stanza"* | Kitchen-dining single room | 1 | | |
| *"Open soggiorno cucina"* | Kitchen living room open area | 1 | | |
| *"Zona giorno bilocale"* | Two-room flat living area | 1 | | |
| *"Soggiorno-cucina"* | Living room-Kitchen | 1 | | |
| *"Soggiorno con angolo cottura"* | Living room with kitchen corner | 1 | | |
| *"Cucina-soggiorno"* | Kitchen-Living room | 1 | | |
| *"Cucina + soggiorno (ambiente unico)"* | Kitchen + Living room (single room) | 1 | | |

**Table 4.** *Cont.*

| Answer Provided by Participants (in Italian in Quotes, the Meaning on the Right) | | Number | Aggregated Answer (Encoding) | Number |
|---|---|---|---|---|
| *"Vano scale"* | Stairs room | 2 | | |
| *"Mansarda"* | Attic | 2 | | |
| *"Corridoio"* | Aisle | 1 | Other (4) | 8 |
| *"Stanza ripostiglio"* | Storage room | 1 | | |
| *"Balcone"* | Balcony | 1 | | |
| *"Zona pranzo, separata da soggiorno e cucina, su suppalco in open space"* | Dining area separate from living room and kitchen on open space | 1 | | |

### 2.3. Data Analysis

The answer to the first question, reported in Section 1.3, analyses the general feeling of the IEQ of homeworkers and whether the different answers are correlated with each other using the Spearman correlation test, which determines the strength and direction of the monotonic relationship between two variables. It was used because it is commonly used to highlight correlations based on data from questionnaires [35–39]. The Spearman correlation test determines the strength and direction of the monotonic relationship between two variables.

The second question relates to the possibility of predicting overall indoor comfort satisfaction, considering a number of selected features. For this purpose, the Python-based libraries scikit-learn [40,41] and XGBoost [42] are used. Among the various supervised learning methods available [43], the appropriate estimators were selected using the flowchart of scikit-learn [44], which is shown in Figure 3.

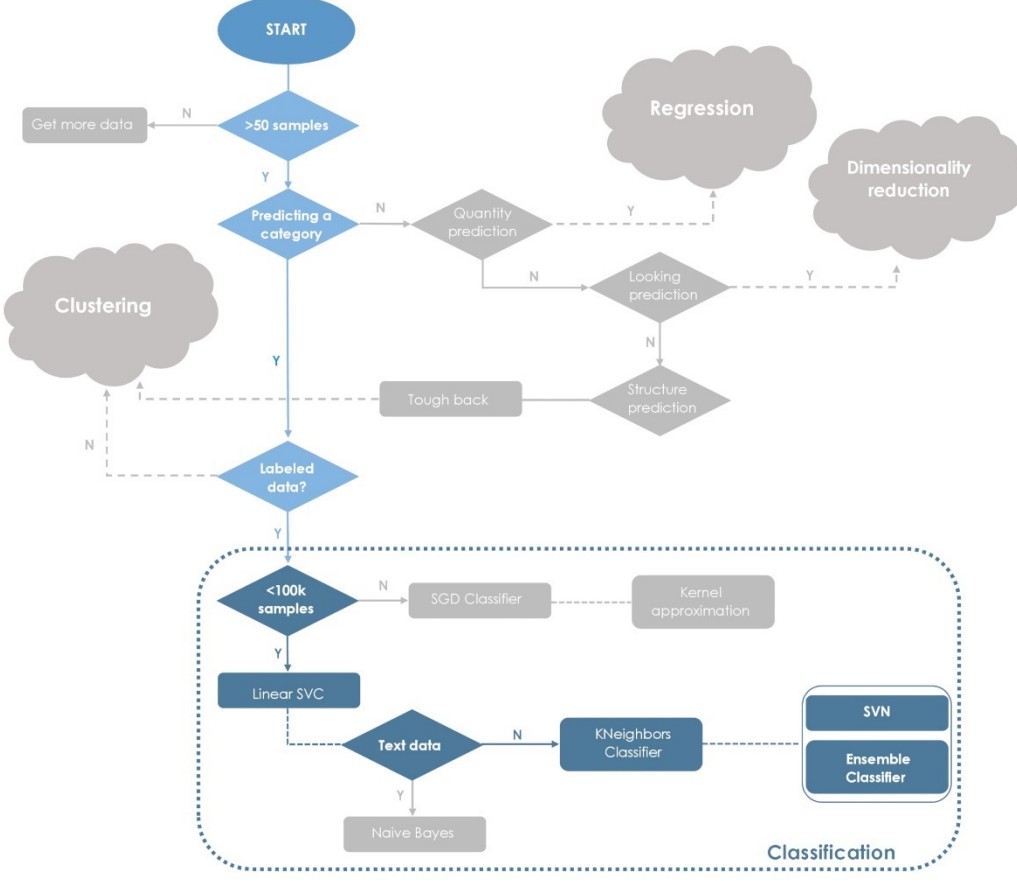

**Figure 3.** The main flowchart used to choose the estimators to use.

Following the flow, the dataset consists of more than 50 and less than 100 k samples and the problem to be solved is typically based on category prediction, with labelled data (e.g., Table 4), used by supervised learning algorithms. This is a typical classification problem, and among the various useful predictors, five different algorithms were selected: LSVC (Linear Support Vector Classifier) [45], kNN (K-Nearest Neighbours) [46], SVM (Support Vector Machines) [47] and two of the most commonly used ensemble classifiers, GBC (Gradient Boosting Classifier) [48] and XGB (eXtreme Gradient Boosting) [49].

The "F1-score" metric was used to evaluate the different algorithms, as it is most commonly used when learning from imbalanced data [50,51]. It is defined as the weighted average of precision (defined as a measure of a classifier's exactness) and recall (considered as the completeness of the classifier). Stratified k-fold cross-validation (number of splits = 10) is used to evaluate the performance of the different algorithms.

The statistical significance of the results is tested using an Analysis Of Variance (ANOVA) ($p$-value < 0.05). The best result in terms of average F1-score is obtained by considering the tuning of the hyperparameters for all the models.

To answer the last question, sentiment analysis of each participant's comments was performed. For this purpose, the VADER (Valence Aware Dictionary and sentiment Reasoner) library [52] is used as it offers substantial benefits over traditional methods of Sentiment Analysis, namely:

- It well fits the "short" texts, such as those used, for example, on social media, reviews of products or services and also in this particular questionnaire;
- It does not require training data, as it uses a combination of a sentiment lexicon as a list of lexical features (e.g., words) which are generally labelled as positive or negative according to their semantic orientation.

VADER reports a compound score for each sentence, which is calculated by adding the values of each word in the lexicon and then normalised between −1 (extremely negative) and +1 (extremely positive). It is a normalised, weighted composite score that gives a one-dimensional measure of the sentiment of a given sentence. Typical thresholds [52] are:

- Positive sentiment: Compound score >= 0.05.
- Neutral sentiment: −0.05 < compound score < 0.05
- Negative sentiment: Compound score <= −0.05.

To validate the automated sentiment analysis, a comparative analysis was performed as usual [52]. All texts were judged by 8 different researchers who did not know the score assigned by VADER choosing between three different categories to describe the general sentiment of each comment: Negative, neutral and positive. An average score was then calculated by considering the overall positive, neutral and negative scores for each evaluator.

## 3. Results and Discussion

The sample of participants consists of 56% females and 44% males (Figure 4a), aged between 26 and 65 years. The breakdown by age group (Figure 5a) is as follows: 39% of participants are in the age range 36–45, 30% in the range 46–55, 19% in the range 26–35 and 12% in the range 56–65. More than 40% are public sector employees (including researchers and administrative staff), about 30% are private sector employees, 14% are university researchers or professors and 12% are freelancers, generally with a high level of education: 74% report having a master's degree and 25% have a high school diploma. When the participants submitted the questionnaire, 75% of them had been WFH for more than 1 month and 22% for more than 2 weeks. Thus, the majority of participants were accustomed to WFH. The survey shows that only 19% of the participants perform their work activities alone, and more than 80% share the spaces with other family members. About 38% share their working life with one or more children (Figure 5b).

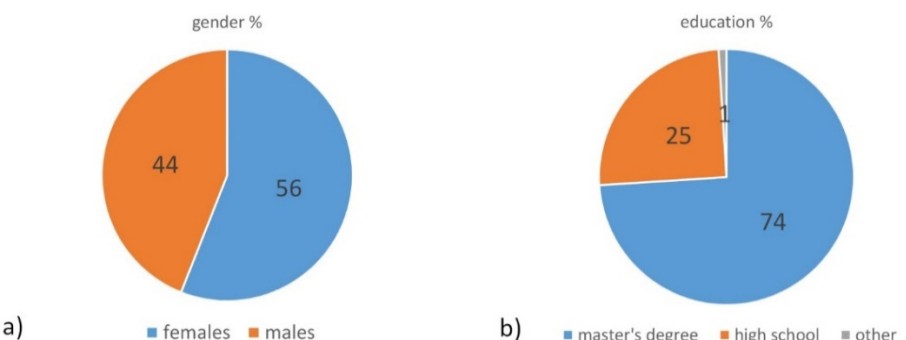

**Figure 4.** Sample distribution: (**a**) Gender; (**b**) education.

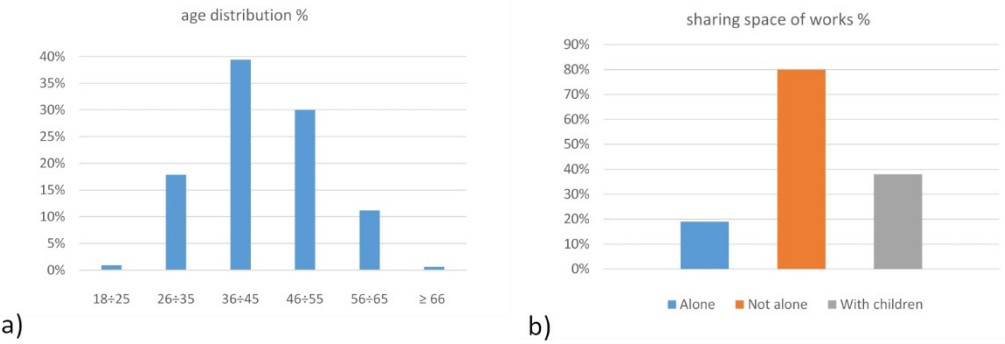

**Figure 5.** Sample distribution: (**a**) Age distribution; (**b**) sharing space of works.

The data collected show that more than 72% of the participants live and work in apartments, while about 45% live in houses built before 1976 (on 30 April 1976, the Italian government enacted Law No. 373 [53], which aims to regulate the performance characteristics of building components, installation, operation and maintenance, and energy use in buildings), which in more than 55% of cases have not undergone major renovations in the last 10 years. About 40% carry out their work activities in the living room and in 78% of the cases less than 2 m from a window. The size of the rooms used is mostly between 10 and 25 m². The time spent in these rooms is between 6 and 9 h for 71% of the respondents. To ensure adequate ventilation, 44% of users leave windows open for more than one hour per day and 55% for more than 10 min. In 49% of cases, users also use window blinds to ensure visual privacy.

*3.1. Goal 1—Descriptive Approach. Question 1: How Do Home Workers Feel the IEQ during Working Activities at Home? Question 2: Are They Satisfied about Their Productivity?*

To answer these two questions, satisfaction, preference, perception and interference of environmental factors in relation to worker productivity were analysed. Satisfaction was evaluated for all considered aspects. For perception and interference, only environmental factors (TC, VC, AQ and IAQ) were considered. For preference, TC, VC and AQ were evaluated. Both TC and VC use a bipolar scale for perception, meaning that the central value (Neutral = 0) is the best perception. It was found that 63% and 78% of the responses on thermal and visual comfort expressed a Neutral perception (mean values of about 0.0 and −0.2, respectively). Furthermore, for a one polar scale, the best perception, which (once) again corresponds to the 0 value, is the least disturbance, i.e., "not smelly" for IAQ and "very quiet" for AQ. The majority of participants, about 74% of the participants, do not perceive any smells, and the mean perception for IAQ is 0.3, between "not smelly" and "slightly smelly", which means that workers' perception for IAQ is also very good. In contrast, responses for AQ are spread across the range, with AQ achieving a mean score of 1.41, ranging between "Quiet" and "Slightly noisy", with 38% of participants rating the acoustic quality of the environment as "Quiet" and 31% as "Slightly noisy". Figure 6 summarises the results for each of the environmental factors considered.

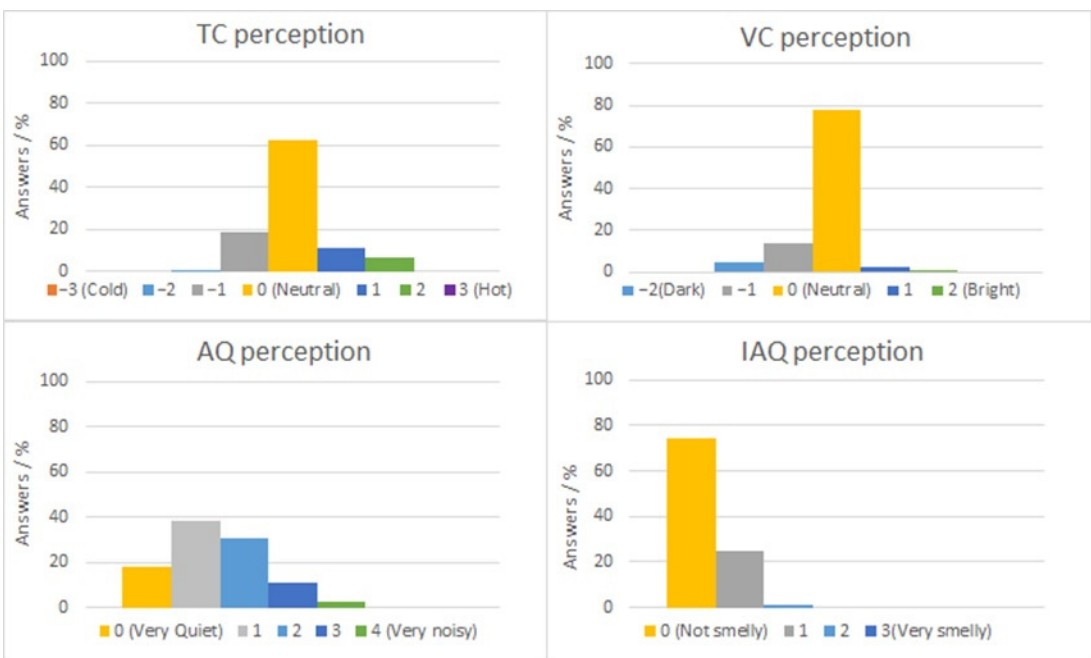

**Figure 6.** Percentage of answers for each Indoor Environmental factors' perception.

Furthermore, the survey investigates how home workers feel about their productivity in relation to the indoor environmental conditions during the lockdown (Figure 7). In general, the distribution of responses shows that participants feel that all indoor conditions (TC, VC, AQ, IAQ) increase their productivity. More than 50% of the participants answer that VC and AQ increase their work productivity. Furthermore, 49% and 45% believe that TC and IAQ improve their work productivity.

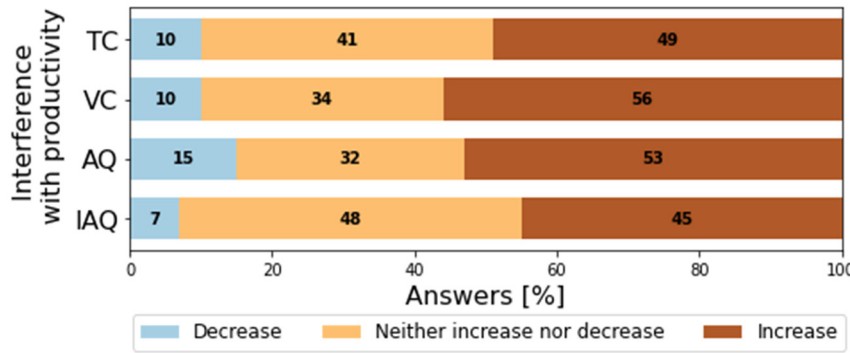

**Figure 7.** Indoor environmental factors interference with productivity.

The responses on thermal and visual preferences show that 288 and 246 of the participants, respectively, indicated that there was no need for change. In terms of acoustic preference, only 166 answered "no change", while 109 responses were "Slightly quieter", representing a total of 49% of participants needing change. This confirms the values from Figure 6 above, where AQ is perceived as not very good (very quiet) and 44% perceived an acoustic quality between "slightly noisy" and "very noisy". Moreover, with regard to AQ, 47% of the participants expressed the "decrease" or "neither increase nor decrease" option with regard to their productivity.

The analysis of expressed satisfaction (Figure 8) shows that more than half of the participants are "Satisfied" and "Very satisfied" with the space, with the lowest value recorded for furnishings and equipment (51%, the sum of "very satisfied" and "satisfied" responses), and the highest value refers to TC (89%, the sum of "very satisfied" and "satisfied" responses). Among the other environmental factors, the lowest satisfaction

is found for acoustics with 68% satisfied, which is consistent with the values of acoustic privacy (67% satisfied) and confirms the perception values of AQ: 13% for "very noisy" and "noisy" considered together and 31% for "slightly noisy".

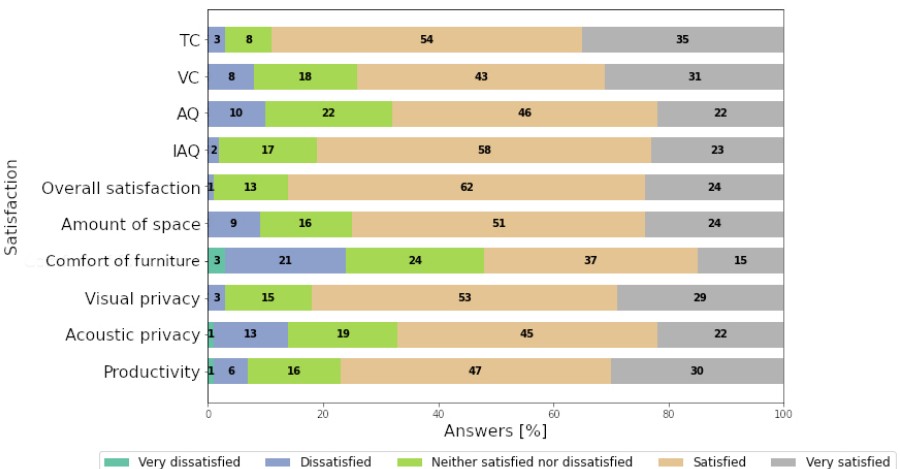

**Figure 8.** Percentage of answers per each aspect of satisfaction.

Considering the score assigned to each possible answer in the satisfaction scale, as indicated in Table 3, the average score of the answers for each of the factors considered is calculated, as shown in Figure 9. The lowest mean score is related to comfort related to furniture and equipment, with a mean score of 0.38, which is close to the "neither satisfied nor dissatisfied" vote. For the aspects related to the IEQ, the mean values for temperature and IAQ are equal to or higher than 1.0 ("Satisfied" category), where temperature = 1.2 and IAQ = 1.0. The mean value of satisfaction for VC and AQ is equal to 0.95 and 0.78, respectively, confirming that AQ is the lowest value. The mean score of overall satisfaction is equal to 1.08.

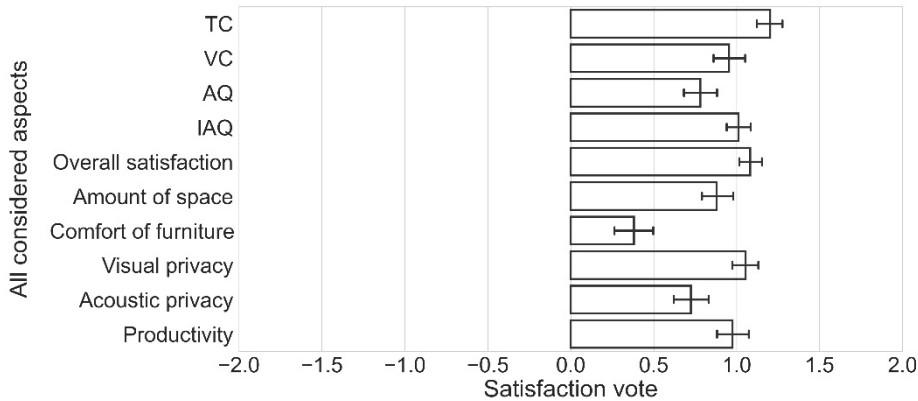

**Figure 9.** Barplot of answers on satisfaction: The length of each rectangle estimates the central tendency for a numeric variable and the error bar provides the indication of the uncertainty (CI = 0.95) around that estimate.

Home workers are exposed to different stimuli from different environmental factors. These stimuli influence the level of satisfaction, perception and preference. However, the description of the combined effects of different environmental factors is performed by analysing a homogeneous rating scale, according to the satisfaction, perception and preference scale defined in EN 16798-1 [54]. In this study, following EN 10551 [55], the satisfaction scale was chosen to assess the level of overall satisfaction of workers. The entire dataset was adjusted for responses related to all preference and perception questions. See Appendix A for a complete list and description of all considered features. Figure 10

correlates the target feature (overall satisfaction) with all other features and shows those with significant results ($p$-value < 0.05).

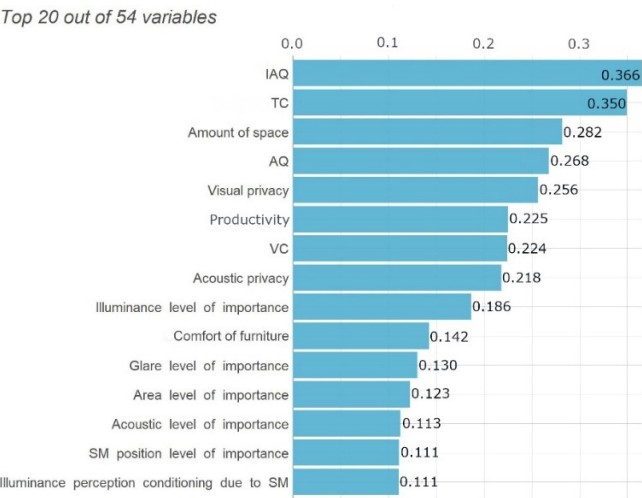

**Figure 10.** Spearman's correlation ($p$-value < 0.05) chart of the overall satisfaction against all other significant features.

In Figure 10, the features are ranked in descending order of correlation. Only 15 features of the whole dataset are ranked. The difference between the displayed features and the setting is due to the very low statistical significance for all other correlations, considering this specific target feature and the significance level set at 0.05. The low statistical significance is not a discriminating factor when performing an ML approach to predict a specific target feature. In [56], it is stated that statistical significance plays little or no role in evaluating predictive performance, while in [57], it is explained and demonstrated in detail that variables with high predictive power may not be significant and, conversely, higher significance does not imply higher predictive power.

As reported in [58], the general idea is to eliminate features that have a poor correlation with the target feature. However, as reported in [59], the fact that some variables are poorly correlated with the target feature does not mean that they cannot contribute to the prediction of the target feature. In the next section, we try to check whether it is possible to use the available data to predict overall satisfaction.

### 3.2. Goal 2—Predictive Model. Question 3: What Are the Most Suitable Algorithms and Influential Features to Predict Workers' Overall Satisfaction?

A total of 54 features are used in the predictive approach with the main goal of identifying which model and features are the best at predicting the overall satisfaction. Appendix A provides a brief description of all features. The features related to perception and preference for all four environmental factors (features 27, 28, 32, 33, 38, 39 and 43 in Appendix A) and note (62 in Appendix A) are excluded from the predictive approach. Firstly, as shown in Figure 8, the "Dissatisfied" option was not considered because it only represents 1% of the total data, unlike the other three options (Very satisfied, Satisfied and Neither satisfied nor dissatisfied) that were considered. Thus, feature selection is performed to reduce overfitting issues, to obtain a simple model that is more explainable by avoiding poor-quality input that can produce noisy outputs.

The main variables that can be used to determine the overall satisfaction are defined considering three main approaches:

- Extra Trees classifier, which is used to estimate the importance of features [60];
- For LSV, GBC and XGB, a recursive feature elimination procedure was performed to identify the number (n) of features useful for the best predictive value and the n most

important features, except for XGB, for which the n most important features were selected considering the built-in feature importance functionality [61];

- For KNN and SVM, a permutation feature importance technique was performed [62] due to the intrinsic nature of these two models that do not support native feature importance scores;

As result, six different lists of features were considered (Table 5).

**Table 5.** Lists of selected features.

| Nr | Feature | List1 | List2 | List3 | List4 | List5 | List6 |
|----|---------|-------|-------|-------|-------|-------|-------|
| 2 | School education level | | ● | | | | |
| 3 | Current occupation | | | | | ● | |
| 5 | Position in WFH (Sitting/Standing/Other) | | ● | | | | |
| 10 | Year of construction of the house | | | | | ● | |
| 11 | Renovation of the residential unit in the last 10 years (Yes/No) | | ● | | | | |
| 12 | Type of renovation intervention performed | | | | | ● | ● |
| 19 | Presence and ease of use of ventilation automatic control system | | ● | | | | |
| 22 | Window type | | | | | ● | |
| 23 | Window exposure | | | | | | ● |
| 26 | TC- Thermal Comfort: temperature satisfaction | ● | ● | | ● | ● | ● |
| 29 | If you generally perceive a discomfort condition, specify which is the main source (you can also choose several options) | | | ● | | ● | |
| 31 | VC -Visual comfort satisfaction | ● | | | | | |
| 34 | Visual discomfort source during WFH | | | | | ● | |
| 35 | Use of artificial lighting during WFH | | | | | ● | |
| 36 | Interference of visual comfort with the ability to perform working activities | | ● | | | | |
| 37 | AQ—Acoustic Quality satisfaction | ● | ● | | | | ● |
| 40 | Interference of acoustic quality with the ability to perform working activities | | | ● | ● | | |
| 41 | Solution adopted to ensure the acoustic privacy | | | | | | |
| 42 | IAQ—IAQ satisfaction | ● | ● | | | | ● |
| 44 | In case of bad air, define the source of odours | | ● | ● | ● | | |
| 45 | Interference of IAQ with the ability to perform working activity | | ● | | | | |
| 47 | Satisfaction with the amount of available space | ● | ● | | | | ● |
| 48 | Satisfaction with the furniture and complements (desk, table, chair) | ● | | | ● | | ● |
| 49 | Satisfaction with the visual privacy | ● | ● | ● | ● | ● | |
| 50 | Satisfaction with the acoustic privacy | ● | ● | ● | ● | ● | ● |
| 51 | Satisfaction with the work productivity | ● | ● | ● | ● | | |
| 58 | Importance given to the illuminance level in defining the overall satisfaction | | | | | | ● |
| 60 | Importance given to the Acoustic Quality in defining the overall satisfaction | | | | | | ● |

Table 6 shows the average F1-score and standard deviation for the different considered models and the list of selected features.

**Table 6.** F1-score for the different models and feature selection. The selected combination is shown in bold.

| List | LSVC AVG ± STD | KNN AVG ± STD | SVM AVG ± STD | GBC AVG ± STD | XGB AVG ± STD |
|---|---|---|---|---|---|
| 1 | 0.694 ± 0.086 | 0.684 ± 0.067 | **0.762 ± 0.064** | 0.748 ± 0.075 | 0.746 ± 0.056 |
| 2 | 0.681 ± 0.090 | 0.688 ± 0.062 | 0.742 ± 0.062 | 0.747 ± 0.092 | 0.736 ± 0.056 |
| 3 | 0.539 ± 0.043 | 0.583 ± 0.066 | 0.617 ± 0.091 | 0.664 ± 0.103 | 0.693 ± 0.042 |
| 4 | 0.663 ±0.073 | 0.672 ± 0.068 | 0.734 ± 0.087 | 0.761 ± 0.088 | 0.758 ± 0.056 |
| 5 | 0.605 ± 0.054 | 0.617 ±0.082 | 0.597 ± 0.094 | 0.709 ± 0.069 | 0.693 ± 0.087 |
| 6 | 0.622 ± 0.078 | 0.616 ± 0.083 | 0.596 ± 0.075 | 0.707 ± 0.078 | 0.708 ±0.063 |

Table 7 shows the range of tuned hyperparameters considered for each algorithm. Among the various techniques available for tuning hyperparameters, the grid search method [63] was used. This method starts with the definition of a search space grid. The grid consists of selected hyperparameter names and values, and the grid search exhaustively searches for the best combination of these given values.

**Table 7.** Hyperparameters tuning range for different algorithms.

| Algorithms | Hyperparameters | Range | Selected |
|---|---|---|---|
| LSVC | Penalty | ['l1', 'l2'] | 'l2' |
| | C_value | [100, 10, 1.0, 0.1, 0.01] | 1.0 |
| KNN | Leaf_size | range(1,10,2) | 1 |
| | n_neighbors | range(1,30,5) | 6 |
| | p_value | [1, 2] | 1 |
| SVM | Kernel | ['poly', 'rbf', 'sigmoid'] | 'rbf' |
| | C_value | [100, 50, 10, 1.0, 0.1, 0.01] | 50 |
| | Gamma | ['auto', 'scale', 100, 10, 1, 0.1, 0.01, 0.001, 0.0001] | 0.01 |
| GBC | Max_depth | range(5,16,2) | 4 |
| | Min_samples_split | range(200,1001,200) | 200 |
| XGB | Learning_rate | [0.05, 0.1, 0.15, 0.2, 0.25, 0.30] | 0.3 |
| | Max_depth | range(3,10,2) | 3 |
| | Min_child_weight | range(1,6,2) | 3 |
| | Gamma | [i/10.0 for i in range(0,5)] | 0 |
| | Colsample_bytree | [0.3, 0.4, 0.5, 0.7] | 1 |

Table 6 shows that the SVM slightly outperforms the GBC and XGB models and has the highest average F1-score (0.762 ± 0.064). Indeed, it is not new that these types of models based on the SVM algorithm perform well and are widely used to predict, for example, users' thermal sensation [64]. The no. 1 list considered to obtain this value is characterised by the features related to the users' absolute comfort level (TC, VC, AQ and IAQ), the home working space ergonomics (Amount_of_space and Comfort_of_furnishing) and the factors that influence the ability to concentrate on work and productivity (Visual_privacy, Acoustic_privacy and Productivity).

It is possible to explain the results of the SVM-based model, by applying the Shapley Additive explanations (SHAP) library, a game-based theoretical approach, thus allowing one to identify the importance of all selected features [65]. Figure 11 shows the distribution of the impacts of each feature on the model output.

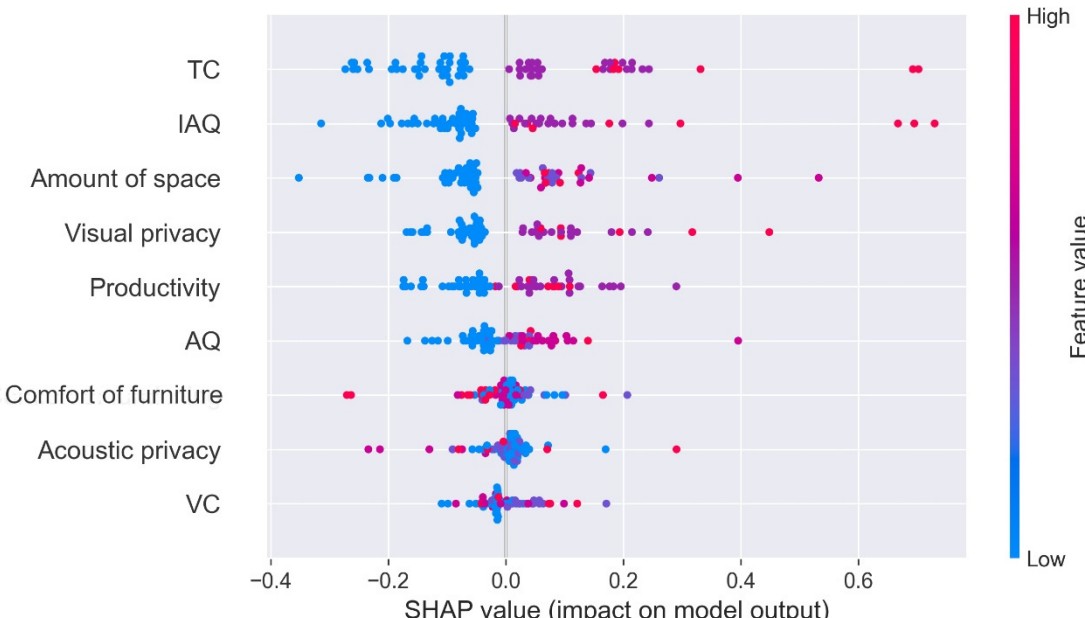

**Figure 11.** SHAP summary plot: High feature values in red; low features values in blue.

The plot variables are classified in descending order of importance. The points, each of which represents a prediction, are distributed along the horizontal axis: The further from the centre of the X-axis (SHAP value = 0.0, which means zero impact on the model output), the greater the impact. The positive SHAP values of the points on the right represent a positive impact on the quality score, while the negative points are associated with an antagonistic impact. Figure 11 shows that among the environmental factors, TC and IAQ have the largest positive impact, while among the other selected features, the most important in terms of predicting overall satisfaction, considering the specific SVM-based model, is the amount of space. For the first six features (TC, IAQ, Amount of space, Visual privacy, Productivity, AQ), it can be seen that higher or neutral satisfaction values generally improve the model prediction.

### 3.3. Goal 3—Sentiment Analysis. Question 4: What Impact Does WFH Have on Workers' Satisfaction?

Only 87 out of 330 participants left comments. Based on the Automated Sentiment Analysis, it can be summarised that 15% of users expressed a positive impact of WFH in their notes, 34 % indicated a neutral impact and 51% stated a negative impact. Moreover, according to the average judgement of eight unaware researchers, 16% of the comments are positive, 16% are neutral and 68% are negative. On the one hand, in both cases, the percentage of positive scores is very low. Therefore, it is important to note that an open-ended question is usually only filled when someone has something to comment on, and that is usually a complaint. On the other hand, in both cases, the negative score is the most relevant. When analysing the main reasons for the negative comments, it can be seen that most of the participants complain about the lack of equipment (chairs, desks, computers, hardware and software) and enough space to work properly and productively in the home environment. In further detail:

- 43% of the sentences expressed negative opinions about WFH because they were not prepared for it due to lack of proper equipment, internet connection, seats and furniture.
- 14% complain about the presence of school-age children due to the closure of schools and distance learning.
- 17% express a negative judgement due to the emergency condition (other roommates in WFH or lack of possibility to go out, etc.).

- 26% express a negative judgement due to the reasons other than those mentioned in the three previous categories.

Even though the sentiment analysis, as stated, only refers to 26% of the sample, the results on furniture and equipment are in line with the result previously found in Section 3.1 on satisfaction, namely that the lowest satisfaction value was recorded for furniture and equipment (52%, the sum of the answers "very satisfied" and "satisfied").

## 4. Conclusions

This study explores the potential of a multi-level approach aimed at assessing the overall users' satisfaction when WFH during the COVID-19 lockdown in Italy. To this end, descriptive, predictive and sentiment analyses of responses to an online questionnaire were conducted.

The predictive approach allowed the SVM-based model to emerge as the most useful for identifying overall satisfaction. In relation to the different satisfaction domains (environmental and other aspects), it has also been shown that subjective and objective features related to the user information and the general data of the building, respectively, have very low importance in determining overall satisfaction during the adaptive WFH experience. With the help of the SHAP library, it was also possible to explain that the productivity satisfaction feature has a very low significance in determining the overall satisfaction of the workers.

At the same time, the descriptive approach highlighted that about half of the participants considered environmental factors as key elements that can positively influence productivity at work. In relation to previous similar studies conducted in office buildings [15,16], the satisfaction vote of people during WFH reaches a positive value on the Likert scale. The lowest vote is related to the "Comfort of furniture", which is due to the lack of work devices in Italian homes. However, people positively perceived the possibility of working in the home environment. It should be noted, however, that these answers could be partly influenced by the particular situation due to the pandemic, and in this sense, further research is needed. These outcomes are partly in contrast to the results obtained when working in the office [15,16], where the acoustics issue (noise level and sound privacy), air quality and temperature are not always perceived positively. At home, everyone has the possibility to adjust their own working environment to achieve the best environmental conditions. In contrast, satisfaction with furniture in the office is higher than at home, highlighting the need to fill this gap with by providing appropriate equipment. Of the four considered factors, VC is considered the most important variable affecting productivity.

In terms of the various environmental factors, AQ has the highest scores for dissatisfaction or neutral satisfaction. The lockdown has significantly changed the acoustic climate both outdoors (drastic reduction of road traffic) [66] and indoors (presence of children or roommates), which may have affected the perceived AQ. The sentiment analysis allows us to confirm this last consideration. As for the other aspects, the comfort of furniture seems to have a negative impact on the satisfaction of the adaptive home worker, with 45% of the users expressing an unsatisfactory or neutral opinion.

Apparently, the results of the sentiment analysis show that workers' perception of WFH was mainly affected by the lack of technology, more than by the disruption of interpersonal working relationships (social aspects), which contradicts numerous previous studies [2,4,5]. In this context, it is possible to distinguish between problems that may occur when WFH is an established practise (reduction of social relations related to work) and an adaptive WFH condition during the emergency period, when particular difficulties may occur (the presence of more than one person in the house, the presence of school-age children), which certainly influenced the responses. It should be noted that the sentiment analysis was conducted with the notes received, which represented only 26% of the total responses. This is certainly a limiting aspect that does not allow generalising the results of the analysis but requires further research.

## 5. Future Works

One of the short-term developments suggested by the team is to resubmit the questionnaire to workers who do their work from home, as public and private sector companies and freelancers may have had the opportunity in the meantime to use a few months of relative quiescence of the virus to make WFH more pragmatic. This would also overcome a possible limitation of the analysis carried out so far, so that other seasons characterised by different temperatures than those recorded for the period under consideration could also be taken into account.

When considering the predictive approach, only significant results were reported in this particular case, which is meaningful in terms of predicting overall satisfaction. Nevertheless, after sharing the newly obtained data again, it will be possible to use them to improve both the predictive performance of the model in terms of the average F1-score and to define new possible predictive scenarios, for example by considering both subjective user data and objective aspects of the occupied buildings and environments, to predict not only overall satisfaction but also satisfaction in terms of productivity. By sharing the questionnaire across Europe, it is also possible to identify differences between countries in terms of satisfaction, perception and preference of IEQ of homeworkers, by applying the multi-level approach described above.

The influence of AQ will be further investigated in a future study by analysing the responses to the acoustic environment questions before and during the lockdown.

**Author Contributions:** F.S. (Francesco Salamone) conceived the idea, wrote the questionnaire and paper with the participation of all authors and performed the predictive approach and automated sentiment analysis on the questionnaire results. All authors participated in determining the judgement of 87 participant feedbacks and in the descriptive analysis. In more detail: A.B., A.D. and M.G. defined all VC-related questions and analysed the corresponding responses; L.D., L.B., I.M. and F.S. (Francesco Salamone) defined all TC-related questions and analysed the corresponding responses; B.B. and F.S. (Francesco Salamone) defined all IAQ-related questions and analysed the corresponding responses; C.S. and F.S. (Fabio Scamoni) defined all AQ-related questions and analysed and interpreted the AQ answers. All authors have read and agreed to the published version of the manuscript.

**Funding:** This research received no external funding.

**Institutional Review Board Statement:** Not applicable.

**Informed Consent Statement:** Informed consent was obtained from all subjects involved in the study.

**Data Availability Statement:** Data are not publicly available due to restrictions regarding the privacy of the participants.

**Conflicts of Interest:** The authors declare no conflict of interest.

## Appendix A

| ID | Feature Label | Description | Question | Possible Answers |
|----|---------------|-------------|----------|------------------|
| 0 | Age_range | Range of age | Please indicate your range of age | 18–25<br>26–35<br>36–45<br>46–55<br>56–65<br>66 or more |
| 1 | Gender | Gender | Please indicate your gender | Male<br>Female<br>I prefer not to answer |
| 2 | Degree | School education level | Please indicate your educational qualification | Primary school diploma<br>Secondary school diploma<br>Graduate<br>Degree (or higher) |

| ID | Feature Label | Description | Question | Possible Answers |
|---|---|---|---|---|
| 3 | Occupation | Current occupation | What is your current occupation? | Public body employee<br>Employee private company<br>Self-employed<br>Dealer<br>Teacher<br>Researcher/University Professor<br>Other (please specify) |
| 4 | Time_in_SW | How long the user has been in WFH | How long have you been in WFH? | From 1 to 2 weeks<br>From 2 weeks to 1 month<br>More than 1 month |
| 5 | Position | Position in WFH (Sitting/Standing/Other) | How do you perform your WFH? | Mainly seated<br>Mainly standing<br>Other (to specify) |
| 6 | How many | Number of people with whom the environment is shared | How many people share the residential unit during your WFH? | Open answer |
| 7 | Children_boolean | Presence/absence of children | Do the persons counted include children? | Yes<br>No |
| 8 | Children_age_range | Range of age of the children | How old are the children? | Less than 6 years old<br>From 6 to 12 years old<br>More than 12 years old |
| 9 | Bld_Type | Type of residential unit | In which type of residential unit are you working? | Detached house<br>Terraced house<br>Flat<br>Other (please specify) |
| 10 | Bld_y | Year of construction of the house | Year of construction of the house | Before 1976<br>Between 1976 and 1991<br>Between 1992 and 2000<br>Between 2000 and 2005<br>After 2005<br>I don't know |
| 11 | Bld_ref_boolean | Renovation of the residential unit in the last 10 years (Yes/No) | Has the residential unit in which you live been renovated in the last 10 years? | Yes<br>No<br>I don't know |
| 12 | Bld_ref_type | Type of renovation intervention performed | What type of renovation intervention has been performed? | Windows substitution<br>External insulation<br>Counter walls<br>Other (please specify) |
| 13 | Env_SM | Room where the WFH has been performed | Identify the room where you perform your smart working activity | Kitchen<br>Living room<br>Bedroom<br>Study room<br>Other (please specify) |
| 14 | Env_area_range | Area of the room | What is the plan size of the identified environment? (if you are not sure, it is suggested to measure the length and width of the room and multiply them to return the correct value) | Less than 10 m$^2$<br>Between 10 and 15 m$^2$<br>Between 15 and 20 m$^2$<br>Between 20 and 25 m$^2$<br>More than 25 m$^2$ |



| ID | Feature Label | Description | Question | Possible Answers |
|----|--------------|-------------|----------|------------------|
| 15 | Env_how_many_hours | Hours spent in the room for WFH | How many hours a day do you usually use this room for WFH? | Less than 6 h<br>Between 6 and 9 h<br>More than 9 h |
| 16 | Therm_usability | Presence and ease of use of thermostat | Please indicate the possible presence and ease of use of the following devices in the residential unit (thermostat) | Easy to use<br>Difficult to use<br>Not present<br>I don't know |
| 17 | Thermostatic valves_usability | Presence and ease of use of thermostatic valves | Please indicate the possible presence and ease of use of the following devices in the residential unit (thermostatic valves) | Easy to use<br>Difficult to use<br>Not present<br>I don't know |
| 18 | Aut_light_usability | Presence and ease of use of lighting automatic control system | Please indicate the possible presence and ease of use of the following devices in the residential unit (lighting automatic control system) | Easy to use<br>Difficult to use<br>Not present<br>I don't know |
| 19 | Air_change_ system_usability | Presence and ease of use of ventilation automatic control system | Please indicate the possible presence and ease of use of the following devices in the residential unit (ventilation automatic control system) | Easy to use<br>Difficult to use<br>Not present<br>I don't know |
| 20 | Windows_open_ range_time | Time with opened windows | Identify how long per day, overall, you leave the windows open | Less than 10 min<br>Between 10 and 30 min<br>Between 30 and 60 min<br>More than 60 min |
| 21 | Env_position_occupied | Position occupied in the room | In relation to the identified environment, what is your usual position? | Near the window or French window (less than 2 m)<br>Far from the window or French window (more than 2 m) |
| 22 | Env_windows_type | Window type | In relation to the identified environment, what kind of window is present? | Single panel window<br>Two or more panels window<br>Single panel French window<br>Two or more panels French window |
| 23 | Env_windows_esposition | Window exposure | What is the window exposure? | North<br>North-East<br>East<br>South-East<br>South<br>South-West<br>West |
| 24 | Visual_privacy _adp_solution | Adapted solution to ensure visual privacy while working | Which solution do you use to ensure the level of visual privacy while working? | Use of the windows awnings<br>Close of the roller shutter<br>No solutions<br>Other |
| 25 | Env_how_many _time_today | Time spent today in the considered environment | How much time have you already spent today in the considered environment? | Less than 1 h<br>Between 1 and 3 h<br>More than 3 h |

| ID | Feature Label | Description | Question | Possible Answers |
|---|---|---|---|---|
| 26 | TC | Thermal Comfort: temperature satisfaction | Are you satisfied with the temperature of the environment? | Very dissatisfied<br>Dissatisfied<br>Neither satisfied nor dissatisfied<br>Satisfied<br>Very satisfied |
| 27 | T_pref | Thermal preference | What would you like the thermal environment to be like? | Cooler<br>No change<br>Warm |
| 28 | T_perc | Thermal perception | How do you perceive the thermal environment? | Cold<br>Cool<br>Slightly cool<br>Neutral<br>Slightly warm<br>Warm<br>Hot |
| 29 | Thermal_discomfort _source | If you generally perceive a discomfort condition, specify which is the main source (you can also choose several options) | What would you like the thermal environment to be like? | Air movements (head, neck, ankles)<br>Temperature too low<br>Temperature too high<br>Air too dry<br>Air too humid<br>Surrounding surfaces too cold or hot<br>Inability to set up control systems properly<br>Inability to adapt clothing to changes in room temperature<br>No discomfort conditions<br>Other (please specify) |
| 30 | TC_interference_ with_SM | Interference of thermal comfort with the ability to perform working activity | How does the thermal comfort of the environment affect your ability to work? | Improves it<br>It is indifferent<br>It gets worse |
| 31 | VC | Visual comfort satisfaction | Are you satisfied with the brightness of the environment where you perform WFH? | Very dissatisfied<br>Dissatisfied<br>Neither satisfied nor dissatisfied<br>Satisfied<br>Very satisfied |
| 32 | V_pref | Visual preference | What would you like the environment brightness to be like? | More dark<br>No change<br>More light |
| 33 | V_perc | Visual_perception | How do you perceive the light of the environment? | Dark<br>Slightly dark<br>Neutral<br>Slightly bright<br>Bright |
| 34 | Visual_discomfort_ source | Visual discomfort source during WFH | If you experience discomfort during the day, please specify the main reason (you can choose more than one option). | Too much daylight<br>Little daylight<br>Too much artificial light<br>Little artificial light<br>Glare<br>No discomfort conditions |

| ID | Feature Label | Description | Question | Possible Answers |
|---|---|---|---|---|
| 35 | Artifical_lighting_use | Use of artificial lighting during WFH | Do you use artificial lighting in WFH? | In the morning<br>In the afternoon<br>In the evening<br>Never |
| 36 | Visual_interference_with_SM | Interference of visual comfort with the ability to perform working activity | How does the visual comfort of the environment affect your ability to work? | Interfere<br>Neutral<br>Enhance |
| 37 | AQ | Acoustic Quality satisfaction | Are you satisfied with the acoustic quality of the environment? | Very dissatisfied<br>Dissatisfied<br>Neither satisfied nor dissatisfied<br>Satisfied<br>Very satisfied |
| 38 | A_vote | | How do you rate the acoustic quality of the environment in which you perform WFH? | Very quiet<br>Quiet<br>Slightly noisy<br>Noisy<br>Very noisy |
| 39 | A_pref | Acoustic preference | How would you like to improve the acoustic quality of the environment in which you perform WFH? | No change<br>Slightly quieter<br>Quieter<br>Much quieter |
| 40 | Acoustic_interference_with_SM | Interference of acoustic quality with the ability to perform working activity | How does the acoustic comfort of the environment affect your ability to perform your work? | Interfere<br>Neutral<br>Enhance |
| 41 | Acoustic_privacy_setting | Solution adopted to ensure the acoustic privacy | What solution do you use to ensure the level of acoustic privacy while working? | Using headphones or similar<br>Closing the door<br>No solution<br>Other (please specify) |
| 42 | IAQ | IAQ satisfaction | Are you satisfied with the indoor air quality? | Very dissatisfied<br>Dissatisfied<br>Neither satisfied nor dissatisfied<br>Satisfied<br>Very satisfied |
| 43 | Odour_perc | | How do you perceive the smell of the environment? | Not smelly<br>Slightly smelly<br>Smelly<br>Very smelly |
| 44 | Odour_source | In case of bad air, define the source of odour | If you perceive a problem with bad air odour, choose which of the following options contributes the most | Kitchen<br>Perfume<br>Products used for cleaning<br>Bad odours generated by instruments (ex. Printer)<br>Bad odours produced by pets<br>No odour problem<br>Other (please specify) |
| 45 | IAQ_interference_with_SM | Interference of IAQ with the ability to perform working activity | How does the air quality level of the environment affect the ability to perform WFH? | Interfere<br>Neutral<br>Enhance |

| ID | Feature Label | Description | Question | Possible Answers |
|---|---|---|---|---|
| 46 | Overall_satisfaction | Overall comfort satisfaction | In general, are you satisfied with the overall comfort level of the environment? | Very dissatisfied<br>Dissatisfied<br>Neither satisfied nor dissatisfied<br>Satisfied<br>Very satisfied |
| 47 | Amount_of_space | Satisfaction with the amount of available space | In relation to the environment used for WFH, are you satisfied with the following aspects? (amount of space) | Very dissatisfied<br>Dissatisfied<br>Neither satisfied nor dissatisfied<br>Satisfied<br>Very satisfied |
| 48 | Comfort_of_furnishing | Satisfaction with the furniture and complements (desk, table, chair) | In relation to the environment used for WFH, are you satisfied with the comfort of furniture? | Very dissatisfied<br>Dissatisfied<br>Neither satisfied nor dissatisfied<br>Satisfied<br>Very satisfied |
| 49 | Visual_privacy | Satisfaction with the visual privacy | In relation to the environment used for WFH, are you satisfied with the following aspects? (Visual privacy) | Very dissatisfied<br>Dissatisfied<br>Neither satisfied nor dissatisfied<br>Satisfied<br>Very satisfied |
| 50 | Acoustic_privacy | Satisfaction with the acoustic privacy | In relation to the environment used for WFH, are you satisfied with the following aspects? (Acoustic privacy) | Very dissatisfied<br>Dissatisfied<br>Neither satisfied nor dissatisfied<br>Satisfied<br>Very satisfied |
| 51 | Productivity | Satisfaction with the work productivity | In relation to the environment used for WFH, are you satisfied with the following aspects? (Productivity) | Very dissatisfied<br>Dissatisfied<br>Neither satisfied nor dissatisfied<br>Satisfied<br>Very satisfied |
| 52 | Area_level of importance | Importance given to the amount of space in defining the overall satisfaction | In defining how satisfied you are with the environment, what importance do you attach to the following aspects? (Available space) | Not at all<br>Not very important<br>Indifferent<br>Very important<br>Extremely important |
| 53 | Furniture_level of importance | Importance given to the comfort of furniture in defining the overall satisfaction | In defining how satisfied you are with the environment, what importance do you attach to the following aspects? (furniture and complements) | Not at all<br>Not very important<br>Indifferent<br>Very important<br>Extremely important |
| 54 | SM_position_level of importance | Importance given to the location of the workstation in defining the overall satisfaction | In defining how satisfied you are with the environment, what importance do you attach to the following aspects? (Location of the workstation) | Not at all<br>Not very important<br>Indifferent<br>Very important<br>Extremely important |

| ID | Feature Label | Description | Question | Possible Answers |
|---|---|---|---|---|
| 55 | Visual_privacy_level of importance | Importance given to the visual privacy in defining the overall satisfaction | In defining how satisfied you are with the environment, what importance do you attach to the following aspects? (Visual privacy) | Not at all<br>Not very important<br>Indifferent<br>Very important<br>Extremely important |
| 56 | Acoustic_privacy_level of importance | Importance given to the acoustic privacy in defining the overall satisfaction | In defining how satisfied you are with the environment, what importance do you attach to the following aspects? (Acoustic privacy) | Not at all<br>Not very important<br>Indifferent<br>Very important<br>Extremely important |
| 57 | Thermal_comf_level of importance | Importance given to the thermal comfort in defining the overall satisfaction | In defining how satisfied you are with the environment, what importance do you attach to the following aspects? (Thermal comfort) | Not at all<br>Not very important<br>Indifferent<br>Very important<br>Extremely important |
| 58 | Illuminance_level of importance | Importance given to the illuminance level in defining the overall satisfaction | In defining how satisfied you are with the environment, what importance do you attach to the following aspects? (Illuminance) | Not at all<br>Not very important<br>Indifferent<br>Very important<br>Extremely important |
| 59 | Glare_level of importance | Importance given to the glare in defining the overall satisfaction | In defining how satisfied you are with the environment, what importance do you attach to the following aspects? (Glare) | Not at all<br>Not very important<br>Indifferent<br>Very important<br>Extremely important |
| 60 | Acoustic_level of importance | Importance given to the Acoustic Quality in defining the overall satisfaction | In defining how satisfied you are with the environment, what importance do you attach to the following aspects? (Acoustic aspects) | Not at all<br>Not very important<br>Indifferent<br>Very important<br>Extremely important |
| 61 | IAQ_level of importance | Importance given to the IAQ in defining the overall satisfaction | In defining how satisfied you are with the environment, what importance do you attach to the following aspects? (Air quality) | Not at all<br>Not very important<br>Indifferent<br>Very important<br>Extremely important |
| 62 | Note | Additional comments | Please provide any additional comments or aspects useful to describe your experience in WFH | Open answer |

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
