# Peer review of "Working from Home in Italy during COVID-19 Lockdown: A Survey to Assess the Indoor Environmental Quality and Productivity"

_buildings, doi:10.3390/buildings11120660_

Round 1

Reviewer 1 Report

It seems an innovating as well as well written work.

Specific comments

  1. Line 92: This literature review is oriented to office environments. Is it possible findings from WFH surveys to be added  ?
  2. Lines 270 – 285: Please describe in more details how the collected data have been used in those models. For example, provide information how the detailed parameters are used (independent vs dependent), any configuration set up, necessary data transformation etc.
  3. Lines 352-353: Please clarify how those calculations have been performed.
  4. Lines 365-367: Please try to be more specific and give more details.
  5. Line 423-424 : Table 6 deserves some interpretation and discussion
  6. Lines 426-435:. The best performance model has been highlighted. What about used models intercomparison ? It would be useful also some discussion concerning other studies using those models.
  7. Lines 452-545: Please give reference(s).

Author Response

It seems an innovating as well as well written work.

We thank the reviewer for the comments and suggestions. We addressed the comments carefully. All changes are in red in the revised version and in double crossed the parts of the text deleted. 

1. Line 92: This literature review is oriented to office environments. Is it possible findings from WFH surveys to be added  ?

We added four new references concerning WFH specific surveys.

2. Lines 270 – 285: Please describe in more details how the collected data have been used in those models. For example, provide information how the detailed parameters are used (independent vs dependent), any configuration set up, necessary data transformation etc.

We added more details in section 2.2 on the data preparation with an example (see new table 4). In section 2.3 we have moved text of the section 3.2, which contains all the information about the metrics, the method of subdivision between train and test, and the methods of selecting the more important features to predict the target feature, the overall satisfaction.

3. Lines 352-353: Please clarify how those calculations have been performed.

The sentence was reformulated by also considering a link to the scores given in Table 3 for each possible answer.

4. Lines 365-367: Please try to be more specific and give more details.

Added a more exhaustive reference.

5. Line 423-424 : Table 6 deserves some interpretation and discussion

Added more details in Table 6 (now table 7) and before Table 6 in the text.

6. Lines 426-435:. The best performance model has been highlighted. What about used models intercomparison ? It would be useful also some discussion concerning other studies using those models.

Table 5 shows a comparison for all models and the set of input feautures considered. Added a brief discussion and a reference in section 3.2 about the use of SVM-based models.

Lines 452-545: Please give reference(s).

An exhaustive reference from which the information was obtained is already provided: [52] C.J. Hutto, E.E. Gilbert, VADER: A Parsimonious Rule-based Model for Sentiment Analysis of Social Media Text. Eighth International Conference on Weblogs and Social Media (ICWSM-14).”, Proc. 8th Int. Conf. Weblogs Soc. Media, ICWSM 2014. (2014).

Reviewer 2 Report

 This study used a survey of 330 residents to assess the impact of working From Home in Italy during COVID-19 lockdown and also the effect of indoor Environmental Quality and productivity

The paper is generally well written. Results make sense and conclusions are based on statistical analysis of the data. I feel the paper is too long and potentially distracting to the readers. Some of the sections such as 1.1 to 1.3 should be consolidated.

The scale comparison in Table 3 seems bit confusing. For eg. it is assumed that “very quiet” is equivalent to “neutral” and that is the preferred condition. This needs to be justifies

Also the period of survey is very crucial and it is not clear which season the surveys were conducted. This will make a huge difference in the perception. For eg. it was mentioned that users leave the window open. If it was during winter and extreme hot days, that will impact the perception. A quick overview of the outdoor weather conditions should be provided and the impact of seasons should be discussed.

English and grammar need to be improved. Some of them are below:

  • In the conclusion section 3rd paragraph – “considered indoor environmental quality factors as an element that can positively influence work productivity” Tis should be corrected.
  • Page 8-Goal 3 – Sentiment analysis. Question 4: Which is the impact of WFH on workers satisfaction-It should be “what” instead of “which”

Author Response

This study used a survey of 330 residents to assess the impact of working From Home in Italy during COVID-19 lockdown and also the effect of indoor Environmental Quality and productivity

The paper is generally well written. Results make sense and conclusions are based on statistical analysis of the data.

We would like to thank you for your feedback. You can find all changes in red in the revised version and in double crossed the parts of the text deleted. 

I feel the paper is too long and potentially distracting to the readers. Some of the sections such as 1.1 to 1.3 should be consolidated.

Sections 1.1 to 1.3 have been revised

The scale comparison in Table 3 seems bit confusing. For eg. it is assumed that “very quiet” is equivalent to “neutral” and that is the preferred condition. This needs to be justifies

Added more details in section 1.3, after bullet point G.

Also the period of survey is very crucial and it is not clear which season the surveys were conducted. This will make a huge difference in the perception. For eg. it was mentioned that users leave the window open. If it was during winter and extreme hot days, that will impact the perception. A quick overview of the outdoor weather conditions should be provided and the impact of seasons should be discussed.

More details on weather data added in section 2.2 and a limitation with a possible future improvement added in section 5.

English and grammar need to be improved. Some of them are below:

  • In the conclusion section 3rd paragraph – “considered indoor environmental quality factors as an element that can positively influence work productivity” Tis should be corrected.
  • Page 8-Goal 3 – Sentiment analysis. Question 4: Which is the impact of WFH on workers satisfaction-It should be “what” instead of “which”

We have revised these sentences and all other parts of the manuscript

Reviewer 3 Report

Overall excellent paper with valuable and inciteful information. However, minor improvements are possible:
Lines 28-29 require citation.
Line 30 unclear which adjectives; consider removing.
Lines 61-63; 71-74; 156-160; 169-171; 491-494; 508-509; 529-536 avoid single sentence paragraphs (possibly by providing additional information or splitting long sentences into several shorter)
Lines 75-79 If impact of IEQ on productivity is well known, it is insufficiently clear why assessment needs to be extended. Consider revising.
Table 1: "objective creative task" unclear, please revise; IT - clarify abbreviation
Lines 156-160 unclear from which reference this information originates; if these are author's own considerations, consider moving them from the introduction section to methods
Line 177: "smart workers" should be defined. Furthermore Question 1 is insufficiently clear.
Line 182: which -> what
Line 185: "next future" - unclear. Furthermore, the whole paragraph is better suited for Methods section.
Lines 288, 300, 317: Sentence should not start with a number.

It would be useful to distinguish Introduction from the Literature review.
The authors should make additional effort to define what they consider as WFH and how is this different from smart working or other terminology they seem to use interchangeably.
Tables 1, 2, 3 formatting should be improved to avoid splitting letters from words in new lines. Distinction between Tables 1 and 2 is insufficiently clear, e.g. why aren't the studies from Table 1 included in Table 2 or vice versa? Additional commentary would be useful on the consensus or lack of consensus within the literature regarding the importance of different factors for workers satisfaction. 
It is insufficiently clear whether Fig 1 is produced by authors or taken from a reference. If produced by authors, more in depth explanation how such data were obtained would be useful to ensure replicability of results. If taken from a reference, citation should be included in the figure caption. Mentioning the source in this figure does not follow the applied/expected citation style used in the paper.
Methods section should describe the used questionnaire in sufficient detail to enable repeatability (e.g. it is insufficiently clear which questions were asked to determine whether participants are working alone, share spaces with family, children, and how long they've been working from home, types of accommodation, year of construction, working location, distance from windows, window opening patterns, use of blinds etc.). All seven sections of the survey should be consistently described, i.e. Sections E, F and G should be also part of the bullet list, like the other four sections (or none of the sections should be in the bullet list). Description for Section F mentioning interference with the working activities does not seem to apply to productivity. It should be further clarified what was the exact question related to productivity that the respondents were asked to answer.
Information about the response rate would be useful.
Font size in Fig 3 should be increased to the size comparable to the surrounding text. Also bold part of the figure caption should be corrected.
It is recommended that some of the demographic statistics described in lines 287-297 are also presented using histograms or pie charts.
It should be clarified why was the year of 1976 relevant in terms of the built environment.
Lines 371-374, 413-417, 421-422, 451-466 are better suited for the methods than results section.
The authors should present evidence that indeed there was a monotonic relationship between the variables they tried to correlate. 
Very little discussion is available about the overall importance and relevance of the presented findings. The authors should make additional effort to better interpret their results, e.g. comparing them to results from other studies prior to the pandemic, discussing similarities or discrepancies to such studies and providing the main outcomes/recommendations in form of a summary. A more in depth description of the best model to predict satisfaction will be also useful, e.g. describe how were hyperparameters in Table 6 selected.
Evidence for the statements in lines 534-536 is necessary.
Very few (if any) references follow a standard referencing style - this has to be rectified!

Author Response

Overall excellent paper with valuable and inciteful information. However, minor improvements are possible:

We would like to thank you for your feedback. Your detailed comments have considerably helped with improving the clarity of the revised manuscript. You can find all changes in red in the revised version and in double crossed the parts of the text deleted. 

Lines 28-29 require citation.

The sentence was revised and a more comprehensive text was reported justified with a reference.

Line 30 unclear which adjectives; consider removing.

we added the adjectives in brackets.

Lines 61-63; 71-74; 156-160; 169-171; 491-494; 508-509; 529-536 avoid single sentence paragraphs (possibly by providing additional information or splitting long sentences into several shorter)

We have revised these sentences and all other parts of the manuscript.

Lines 75-79 If impact of IEQ on productivity is well known, it is insufficiently clear why assessment needs to be extended. Consider revising.

The difference between previous works and the current one was added at the end of section 1.1.

Table 1: "objective creative task" unclear, please revise; IT - clarify abbreviation

Although "objective creative task" was mentioned in the reference, we opted for "creative performance." "IT" stands for Information Technologies: this was clarified in Table 1.

Lines 156-160 unclear from which reference this information originates; if these are author's own considerations, consider moving them from the introduction section to methods

As suggested, this sentence has been moved in the "Methods" section and modified.

Line 177: "smart workers" should be defined. Furthermore Question 1 is insufficiently clear.

We revised all part of the text where the term "smart working" or "smart worker" was considerd, except for figure 1, becasue, as stated in the text, and in the caption of the figure, "smart working" is commonly used in Italy to refer to WFH.

Line 182: which -> what

Modified as requested.

Line 185: "next future" - unclear. Furthermore, the whole paragraph is better suited for Methods section.

The sentence was deleted in the revised version of the manuscript.

Lines 288, 300, 317: Sentence should not start with a number.

The sentence have been modified.

It would be useful to distinguish Introduction from the Literature review.

In our opinion, in addition to introducing the study, the introductory section should also illustrate the previous relevant experiences with the subject of the article in a unique discourse. In this case, in order to facilitate the reading of the section, we decide to maintain the subdivision of the contents into specific subsections.

The authors should make additional effort to define what they consider as WFH and how is this different from smart working or other terminology they seem to use interchangeably.

In all the text of the revised manuscript we used the term WFH, except for the Figure 1, where you can read "smart working" because, as stated in the text, and in the caption of the figure, "smart working" is commonly used in Italy to refer to WFH.
Tables 1, 2, 3 formatting should be improved to avoid splitting letters from words in new lines. Distinction between Tables 1 and 2 is insufficiently clear, e.g. why aren't the studies from Table 1 included in Table 2 or vice versa? Additional commentary would be useful on the consensus or lack of consensus within the literature regarding the importance of different factors for workers satisfaction. 

A sentence has been added in the introduction to justify the main difference between Table 1 and Table 2. Added four new references to WFH-specific surveys.

It is insufficiently clear whether Fig 1 is produced by authors or taken from a reference. If produced by authors, more in depth explanation how such data were obtained would be useful to ensure replicability of results. If taken from a reference, citation should be included in the figure caption. Mentioning the source in this figure does not follow the applied/expected citation style used in the paper.

The figure 1 is produced by authors using the pytrends library application considering the list of keywords given in the legend, no specific category, time frame between 2019-04-15 and 2020-10-15, geo="IT", no group. We added also the reference no. 24 for more details.

Methods section should describe the used questionnaire in sufficient detail to enable repeatability (e.g. it is insufficiently clear which questions were asked to determine whether participants are working alone, share spaces with family, children, and how long they've been working from home, types of accommodation, year of construction, working location, distance from windows, window opening patterns, use of blinds etc.). All seven sections of the survey should be consistently described, i.e. Sections E, F and G should be also part of the bullet list, like the other four sections (or none of the sections should be in the bullet list). Description for Section F mentioning interference with the working activities does not seem to apply to productivity. It should be further clarified what was the exact question related to productivity that the respondents were asked to answer.
Information about the response rate would be useful.

Section E, F and G are now part of the bullet list. More details about the questionnaire are reported in Appendix A.

Font size in Fig 3 should be increased to the size comparable to the surrounding text. Also bold part of the figure caption should be corrected.

Figure 3 has been redrawn to overcome possible copyright issues.

It is recommended that some of the demographic statistics described in lines 287-297 are also presented using histograms or pie charts.

Considering your reccomendation, two new figures (4 and 5) are introduced in the new revised version of the manuscript.

It should be clarified why was the year of 1976 relevant in terms of the built environment.

A reference to the first law issued by the Italian government in terms of energy containment for buildings has been added.

Lines 371-374, 413-417, 421-422, 451-466 are better suited for the methods than results section.

All parts are moved to the method section as required.

The authors should present evidence that indeed there was a monotonic relationship between the variables they tried to correlate. 

Effectively, the input features are low-correlated with the target feature. However, low correlation could not imply low power prediction as stated in reference 59. Added more details at the end of section 3.1 with two references.

Very little discussion is available about the overall importance and relevance of the presented findings. The authors should make additional effort to better interpret their results, e.g. comparing them to results from other studies prior to the pandemic, discussing similarities or discrepancies to such studies and providing the main outcomes/recommendations in form of a summary.

Added more details in the conclusions.

A more in depth description of the best model to predict satisfaction will be also useful, e.g. describe how were hyperparameters in Table 6 selected.

Added more details in section 3.2 and a column in table 6 (now table 7).

Evidence for the statements in lines 534-536 is necessary.

The first part of section 5 was revised and this sentence was removed.

Very few (if any) references follow a standard referencing style - this has to be rectified!

The references were revised in accordance with Buildings-MDPI template.

Round 2

Reviewer 2 Report

The paper has improved from the previous version.

However, there are a number of typographical errors which need to be fixed. Some of them are listed below. There are many others which need to be thoroughly checked.

In page 11- “The questionnaire was structure to better understand the potential impact of the sudden massive use of WFH on people, can be undestood by considering the following  inluencing factos:”

Page 17 “VADER reports for each sentence a compound score calculated by summing the valence scores of”

Conclusion “Linkert scale”

“positive felling”

“in controst whit”

Author Response

We have fixed the typographical errors. Thank you for your comments.